# Lung Epithelial Cell Line Immune Responses to *Pneumocystis*

**DOI:** 10.3390/jof9070729

**Published:** 2023-07-06

**Authors:** Theodore J. Kottom, Eva M. Carmona, Andrew H. Limper

**Affiliations:** Thoracic Diseases Research Unit, Departments of Medicine and Biochemistry, Mayo Clinic, Rochester, MN 55905, USA; carmona.eva@mayo.edu (E.M.C.); limper.andrew@mayo.edu (A.H.L.)

**Keywords:** β-glucans, CLR (C-type lectin), inflammation, *Pneumocystis*

## Abstract

*Pneumocystis* sp. are fungal pathogens and members of the Ascomycota phylum. Immunocompetent individuals can readily eliminate the fungus, whereas immunocompromised individuals can develop *Pneumocystis jirovecii* pneumonia (PJP). Currently, over 500,000 cases occur worldwide, and the organism is listed on the recently released WHO fungal priority pathogens list. Overall, the number of PJP cases over the last few decades in developed countries with the use of highly effective antiretroviral therapy has decreased, but the cases of non-HIV individuals using immunosuppressive therapies have significantly increased. Even with relatively effective current anti-*Pneumocystis* therapies, the mortality rate remains 30–60% in non-HIV patients and 10–20% during initial episodes of PJP in HIV/AIDS patients. Although the role of alveolar macrophages is well studied and established, there is also well-established and emerging evidence regarding the role of epithelial cells in the immune response to fungi. This mini review provides a brief overview summarizing the innate immune response of the lung epithelium and various continuously cultured mammalian cell lines to *Pneumocystis*.

## 1. Introduction

*Pneumocystis* is a fungus and a member of Ascomycota [1]. They are also related to *Saccharomyces cerevisiae* and *Schizosaccharomyces pombe* [2]. Furthermore, it has been found in the lungs of several mammalian species [3,4,5]. In fact, it has been proposed that these fungi may have the potential to infect all mammals [6,7]. To date, the complete life cycle model of the organism is unknown (due to its inability to be grown in continuous axenic cultures [8]). These culture challenges in the growth of *Pneumocystis* outside of the mammalian lung have been nicely summarized by Cushion et al. [9]. The focus of this present minireview is the summary of the innate immune response of the host lung epithelium and continuous epithelial cell lines to *Pneumocystis*.

## 2. Lung Epithelium Cell Types

The lung epithelium is one of the first contact points in the mammalian host for *Pneumocystis* and is critical in the pathogenesis of Pneumocystis pneumonia (PCP). Specifically, lung-epithelium-cell-type interactions with *Pneumocystis* include alveolar type 1 (AT1) and type 2 (AT2), Club (formerly called Clara cells) [10] cells, and goblet cells.

### 2.1. Alveolar Type 1 (AT1)

Alveolar type 1 (AT1) cells are squamous cells that account for 95% of the alveolar surface area. Extremely thin in nature (0.1 µM thick) [11], they lie in close proximity to lung endothelial cells, allowing efficient gas exchange [12]. The AT1 cell interactions with *Pneumocystis* have been known for some time now. Lanken et al. used electron microscopy in a rat PCP model to trace the course of infection. He reported the trophic form of the fungus bound to the AT1 cells, and, after 8 days of infection, the host cells remained intact and no damage as a result of inflammation was apparent. After one month of PCP, the Type 1 pneumocytes in contact with the trophic forms displayed significant necrosis. Additionally, an infiltration of AT2 cells was noted, presumably to replace injured/dead AT1 cells. Interestingly, in the same timeframe, the alveolar-capillary membrane remained intact [13]. Yoshida et al. also noted, through electron microscopy studies in the early stages of PCP infections in rats, the alignment of numerous tropic forms with AT1 cells. In addition, they noted that AT1 cell cytoplasm often protruded from the pneumocyte, and thus hypothesized that this close association (often referred to as interdigitating [1]) was important for nutrient acquisition from the host [14], which is a hypothesis with growing support [9,15,16]. Others have also reported, in ultrathin paraffin and plastic-embedded lung tissue from PCP infected mice, the tight associations of trophic as well as cyst forms with the AT1 cell type. In exuberant PCP infections, these researchers reported an abundance of both fungal life forms along with host cell debris and an infiltration of myeloid cells, including macrophages and neutrophils [17]. Beck et al. were one of the first labs to culture alveolar epithelial cells (AECs) to study AEC/*Pneumocystis* interactions in vitro. They showed that fluorescently labeled *Pneumocystis carinii* cultured with alveolar epithelial cells over 3 days remained metabolically active, as did the lung cell substrate. They noted by immunofluorescent staining that AEC cell E-cadherin and occludin, as well as the measures of transepithelial resistance, were unchanged. This led to the hypothesis that the influx of inflammatory cells was needed for epithelial cell damage [18]. In 2003, Limper et al. showed, for the first time, the role of AECs in the innate immune response to *Pneumocystis*. Briefly, stimulating AT1 cells in vitro with *Pneumocystis* β-glucans stimulated the production of macrophage inflammatory protein-2 (MIP-2) at both the mRNA and protein level via a lactosylceramide-dependent mechanism. The addition of a blocking antibody to glycosphingolipid lactosylceramide (CDw17) resulted in a significant reduction in the MIP-2 secretion in *Pneumocystis*-β-glucans-stimulated AT1 cells. Additionally, these authors demonstrated that targeting de novo glycosphingolipid biosynthesis with a specific inhibitor significantly reduced MIP-2 AT1 secretion [19]. Later, this same lab linked p65 NF-κB, via the observation of the significant nuclear translocation of the transcription factor in this cell type, with *Pneumocystis* β-glucans stimulation. Inhibiting NF-κB also resulted in a subsequent downstream of MIP-2 and tumor necrosis factor alpha (TNF-α) mRNA production [20]. Evans et al. concluded these studies in 2012 by showing that incubating AECs with glycosphingolipids and cholesterol microdomain inhibitors resulted in a significant reduction in the expression of TNF-α and MIP-2. They also showed that, by various microscopy analyses, the carbohydrates were internalized by microdomain-mediated mechanisms. The authors’ data demonstrated, for the first time, the vital role the AEC microdomains plays in the innate response to *Pneumocystis*, and that microdomain-targeted therapeutic intervention might be a promising avenue through which to treat PCP-associated lung inflammation [21]. In 2014, it was demonstrated that the specific deletion of the inhibitor of kB Kinase 2 (IKK2) from mouse lung epithelial cells led to a delayed onset of Th17 and B cell responses in the PCP lung. Additionally, the significant delay in the clearance of *Pneumocystis* organisms in the IKK2 epithelial cell knockout in the lung when compared to wildtypes was noted. This study was the first to link the importance of the host lung epithelial cell response to *Pneumocystis* with the regulation effects of adaptive immune responses [22]. 

### 2.2. Alveolar Type 2 (AT2)

Alveolar type 2 (AT2) are cuboidal surfactant secreting cells that aid in surface tension reduction and prevent alveolar collapse [23]. As before with the first reports of *Pneumocystis* AT1 cell interactions, electron microscopy was utilized to examine fungal + type II pneumocyte interactions. Yoshida et al. reported that as noted above, unlike with AT1 cells where numerous trophic forms were found bound to these cells, these life forms were absent from AT2 cells [14]. Another electron microscopy study around that time determined that there were indeed some trophic life forms bound to AT2 cells, but these were much less frequent then noted with AT1 cells [24]. Pesanti provided the first in vitro analysis of AT2 cells and *Pneumocystis*, linking the possibility of this epithelial cell type having an innate immune response to the fungus. He separated *Pneumocystis* organisms from the AT2 cells using transwell inserts. TNF-α was added to the wells, and after incubating the co-cultures for 16–18 h, he observed significant dose-responsive decreases in *Pneumocystis* viability. This was measured by the ^14^CO_2_ release generated by the fungi. The author did not determine the exact cause of the fungal killing by the AT2 cells but stated that the “secretions” of the AT2 cells may be responsible in this co-culture system. This study demonstrated, for the first time, that AT2 cells mount a host immune response to the organism. [25]. The first observation of a known innate immune response to *Pneumocystis* in AT2 cells was reported in 2005. It was shown that when *Pneumocystis murina* was incubated with AT2 cells (purity determined by modified Papanicolaou staining and intracellular staining for human surfactant protein C (SPC)), a detectable kB binding activity was noted, suggesting NF-kB signaling in this cell type [26]. Next, this same lab group demonstrated that monocyte chemotactic protein-1 (MCP-1) was shown to be co-localized to the lungs of a PCP mouse model with AT2 cells (purity determined by SPC staining). Isolated mouse AT2 cells cultured in vitro and incubated with *Pneumocystis* activated JNK were followed by MCP-1 protein release. The specificity of this activation was demonstrated with the pharmacological inhibition of this pathway [27]. These studies with AT2 cells were important initial findings that demonstrated the potential contribution of this cell type to immune-mediated lung injury in PCP.

### 2.3. Club Cells

Club cells, first described in 1881 by Kolliker [28] are cuboidal in shape, non-ciliated, and located in the terminal bronchioles. Their primary functions are secretion (source of club cell secretory protein (CCSP)), barrier integrity, and metabolism [29]. An eloquent in vivo study by Méndez et al. described how an organism can induce the overproduction of mucin in the distal airways of a rat PCP infection. They noted an increase in mucin production in PCP rat lungs compared to the respective control animals at 60 and 80 days of age. Conversely, they noted a significant decrease in the club cell marker CC10 after 80 days in the PCP infected rats. Furthermore, club cells can go through the process of trans differentiation to goblet cells via the activation of the Notch pathway [30]. Although these researchers noted no activation of this pathway in the club cells that were in the presence of *Pneumocystis*, they did report transformation of club to goblet cells in a *Pneumocystis* infection [31].

## 3. Continuous Cultured Lung Epithelial Cell Lines (Table 1)

### 3.1. A549

The A549 cell line is a human-lung-derived adenocarcinoma alveolar-basal epithelial cell that was isolated in 1972 [32]. Highly characterized, they are used as a model for primary AT2 cells [33]. A number of studies have shown the importance of A549 interactions with *Pneumocystis* in the context of attachment and/or organism proliferation/viability [34,35,36,37,38,39,40,41]. This cell line is the most studied in regard to *Pneumocystis* lung epithelial cell line innate immune responses. Limper et al. was the first to show that the media from alveolar macrophages incubated with *Pneumocystis* enhanced the intercellular adhesion molecule-1 (ICAM-1) expression in A549 cells. The addition of a blocking antibody to TNF-α inhibited this response. Next, this paper reported that when *Pneumocystis* was directly cultured on this cell line, ICAM-1 was also significantly increased. Overall, the ICAM-1 secretion by A549 was noted to be less with the fungus on the cell line alone versus the supernatant from the macrophages being exposed to the organisms [42].

Pottratz et al. was the next to show the inflammatory potential of *Pneumocystis* on A549 cells. They showed that, as soon as after 2 h incubation of A549 cells with *Pneumocystis*, interleukin 6 (IL-6) was detectable in culture media and that this synthesis continued for 48 h. Fascinatingly, as IL-6 levels increased in the cultured environment, the production of fibronectin increased, with the authors suggesting that IL-6 is responsible for this increase in the matrix protein. Additionally, the overall organism attachment also increased, indicating potential pathogen modulation of the host epithelial cells that promote organism proliferation [43]. Our lab has previously demonstrated that PCINT1, a potential receptor for *Pneumocystis*, binds to fibronectin and may play a role in organism attachment and proliferation in vivo [44]. Furthermore, these same researchers previously demonstrated that *Pneumocystis* could modify lung epithelial fibronectin-binding integrins in cultured cells [37]. 

Subsequently, it was shown that purified *Pneumocystis* major surface glycoprotein (Msg), when applied to this cell line, could enhance the IL-8 response of A549 cells from 4 to 24 h post treatment. Competition studies with the co-incubation of yeast mannan or fungal β-glucans significantly reduced the release of IL-8, suggesting the presence of carbohydrates epitope(s) that are present in the Msg fraction. The data presented in this manuscript are the first to link the alterations of the A549 immune response to the predominant surface glycoprotein of the organism [45]. 

Four years later, this same author confirmed the binding of Msg via A549 mannose and glucan receptors, and that the administration of glucocorticosteriods can dampen the IL-8 release from this cell line [46]. Liu et al. further showed the usefulness of the A549 cell line in studying the lung epithelial response to *Pneumocystis*. Through the use of siRNA technology, they inhibited the expression of A549 *MUC1* mRNA expression. MUC1, a member of cell surface mucins, is expressed by epithelial cells and protects the apical cell membrane [47]. The inhibition of A549 *MUC1* RNA resulted in decreased epithelial cell binding, as well as downstream ERK1/2 phosphorylation [48]. 

Lastly, Kottom et al. utilized oligonucleotide microarrays to survey the A549 transcriptome after incubation of the cell line with *Pneumocystis* for 3 h. Mixed *Pneumocystis* life forms were allowed to incubate in transwell inserts above the cell line or directly on the epithelial cell line itself. Additionally, separated cyst or trophic life forms directly in contact with the cell line were analyzed. Transcript abundance from approximately 18,400 human genes was analyzed. Depending on the *Pneumocystis* life forms in contact with the A549 cell, an increase in total mRNA fold change (> 4-fold) was noted with the mixed life form populations showing greater fold changes than the isolated cyst and trophic populations or the mixed populations in the transwell inserts. This study confirmed the previously described upregulated immune response proteins induced in the presence of *Pneumocystis*, including ICAM-1, IL-8, and MCP-1 [19,42,45,46]. In addition, a number of additional innate immune response transcripts were identified upon organism contact with A549 cells, including superoxide dismutase precursor (*SOD2*), TNF-α induced protein 2, 3 (*TNFAIP2/3*), *CXCL2*, *GM-CSF*, *CCL20*, and *IL-1a*. Regarding specific genes that were upregulated or downregulated (4-fold) in cyst life forms, a total of 26 genes were identified. Specific transcripts altered in A549 cells upon contact with this life form include *ID2*, a helix–loop–helix (HLH) transcription regulator that plays a crucial role in cell growth and differentiation [49], and superoxide dismutase precursor 2 (*SOD2*), which is involved in cellular cytokine responses [50]. Recently, *SOD2* has been associated with *Scedosporium* spp., the second most common filamentous fungus found in cystic fibrosis patients [51]. Genes that were upregulated upon trophic/A549 contact include an unknown gene/protein product (AL556438) and an inhibitory of DNA binding (*ID-1H*) transcript, which may have roles in cell growth, senescence, and differentiation [49]. Lastly, mitogen-activated protein kinase (MAPK) phosphatase 1 (*MKP-1*) was significantly induced upon its binding to the A549 epithelial cell line. *MKP-1* upregulation was also observed in mice with systemic *Candida albicans* infection. In these infected animals, reduced MKP-1 protein levels due to glutathione reductase (Gsr^−/−^) knock-in resulted in elevated p38 and JNK activity [52]. It is interesting to hypothesize that *Pneumocystis* trophic forms may upregulate this transcript to enhance lung MAPK phosphatase activity, thereby creating an anti-inflammatory environment in the lung to promote organism survival or proliferation.

This survey report provides an initial template for studying these and many other transcripts that are important in both mixed populations as well as in the isolated life forms of *Pneumocystis* and lung epithelial innate immune responses [53].

### 3.2. HAEo^−^

1HAEo^−^ is a human airway epithelial cell line that is immortalized with an aberrant SV40 origin of replication. Physiologically, it retains many of the characteristics of normal airway epithelia, such as tight junction formation and cytokeratin expression [54]. This cell line was used to show that when *Pneumocystis* β-glucans were applied to the airway cells, they release IL-8 in a dose-dependent fashion, and this response was also calcium-dependent. Furthermore, through the use of a transfected reporter assay, IL-8 release was shown to be mediated by NF-kB/AP-1 which activates the downstream MAPK, ERK1/2 pathways. These authors concluded that *Pneumocystis* binding to human airway epithelial cells causes IL-8 secretion, which may help contribute to the early neutrophil immune response in PCP [55]. Currently, it remains unknown if this cell line would display similar cytokine responses to normal non-immortalized human airway epithelial cells. 

### 3.3. Murine Lung Epithelial Cell Line 12 (MLE-12)

MLE-12 cells have certain features of normal type II airway epithelial cells [56], including the expression of lung phospholipids and surfactant proteins [57]. This line has been used in the past to study epithelial cell lung interactions with pathogens, such as *Pseudomonas aeruginosa* and influenza A [58,59]. After demonstrating that this cell line could be used for studying the binding kinetics of *Pneumocystis*, these researchers showed that the binding of *Pneumocystis* β-glucans to this cell line, can also result in the phosphorylation of the EphA2 receptor itself, resulting in IL-6 cytokine release [60].

### 3.4. Murine Lung Epithelial Cell Line 12 (MLE-15)

The MLE-15 lung epithelial cell line has the characteristics of distal bronchiolar and alveolar epithelial cells with many AT2 phenotypes, including surfactant protein synthesis/secretion and phospholipid secretion [57]. Wang et al. reported that, in this cell line, a timed and *Pneumocystis* organism dose-dependent increase in macrophage inflammatory protein-2 (MIP-2) was noted. Next, via transient transfection experiments with consensus kB binding sequences, *Pneumocystis* organisms were shown to activate the NF-kB signaling pathway. The data combined suggest that *Pneumocystis* can affect epithelial cell gene expression, promoting an inflammatory environment in the PCP lung [26].

**Table 1 jof-09-00729-t001:** List of lung epithelial cell lines used in immune response studies to *Pneumocystis*.

Cell Line	Source	Immune Response	Reference
A549	Lung explant culture, epithelial-like (human)	CCL20 CXCL2 GM-CSF ICAM-1 IL-1a IL-6 IL-8 MCP-1 TNF-α TNFA IP2/3	[19,42,43,45,46,48,53]
1HAEo^−^	Airway epithelial (human)	IL-8	[55]
MLE-12	Airway epithelial (mouse)	IL-6	[60]
MLE-15	Airway epithelial (mouse)	MIP-2	[26]

## 4. Concluding Remarks

As noted above, lung epithelial cells are some of the first cell types encountered by *Pneumocystis* in the host lung, and they provide vital innate immune responses to the fungus (Figure 1). Although, at baseline, lung epithelial cells express few or no C-type lectin receptors (CLRs) [53,60,61] that bind fungal mannoproteins or the embedded carbohydrates that line the fungal cell wall, they do also possess other receptors with different immune signaling pathways, such as the EphA2 receptor kinase pathway recognizing fungal β-glucans, which leads to Stat3 and MAPK phosphorylation, as well as the subsequent release of alarmins, cytokines, and chemokines [60,62]. It has been reported that there are more pattern recognition receptors for fungi than any other organisms [63]. As reported above, our understanding of the roles of specific bona fide epithelial receptors to the organism and the downstream inflammatory host response to *Pneumocystis* is still in its infancy. As a result, the role of lung epithelial cells immune responses to *Pneumocystis* is lacking. With the recent progress and contributions of new technologies, such as single cell RNA transcriptomics in fungal pathogenesis [64,65,66,67], as well as exciting preliminary work via the use of lung organoids for the culture and potential propagation of *Pneumocystis*, these new methods may be extremely beneficial for understanding the role of host epithelial cells in the immune response to the fungus. Tisdale-Macioce et al. recently reported that, through the injection of these cultured organoids with *P. murina*, the fungal cell wall component Msg was observed in the organoids themselves. Additionally, AT1- and AT2-cell-specific staining were noted within the lung organoids [68]. This system better represents the host lung environment and is, perhaps, therefore an excellent resource for the understanding of *Pneumocystis*/lung epithelium interactions leading to proinflammatory responses. In addition, this study has also potentially uncovered therapeutic targets for dampening the detrimental inflammation in the lungs of those with PCP.

## Figures and Tables

**Figure 1 jof-09-00729-f001:**
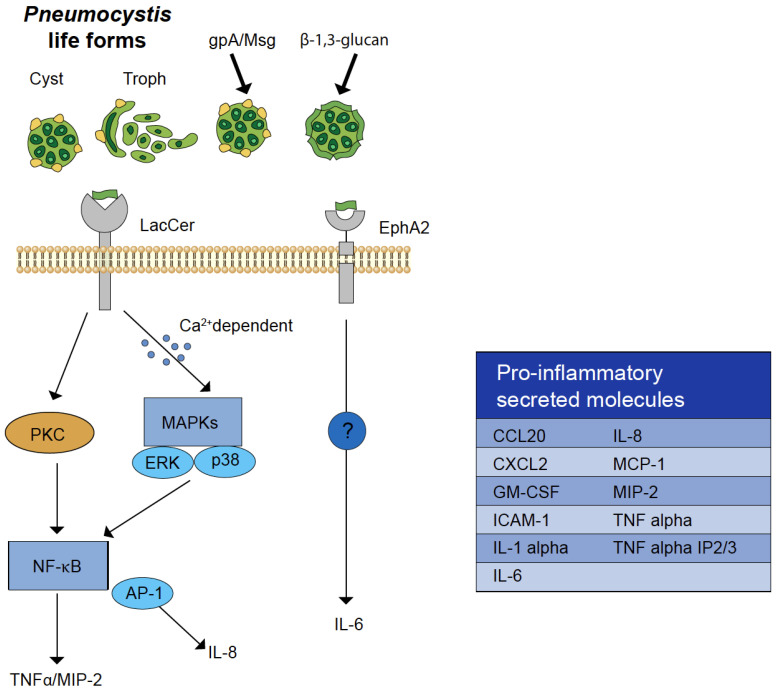
Graphical representation of *Pneumocystis* β-glucan-mediated lung epithelial immunity. EphA2 and lactosylceramide (LacCer) functions as receptors for carbohydrates. Schematic illustration adapted from Wang et al. [69] of the most characterized lung epithelial cell carbohydrate recognition receptors for the fungal organism and brief description of the host response. Both major surface glycoprotein (gpA/Msg) and *Pneumocystis* β-glucan epitopes are shown with black arrows and are host receptor ligands on the *Pneumocystis* cell surface. Mechanism(s) for the IL-6 secretion via EphA2 receptor/*Pneumocystis* β-glucans engagement in airway epithelial cells is unknown.

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
