# Peer review of "Lung Epithelial Cell Line Immune Responses to Pneumocystis"

_jof, 2023, doi:10.3390/jof9070729_

Round 1

Reviewer 1 Report

This is a well written mini review summarizing the innate immune responses of the host lung epithelium and of various continuous mammalian epithelial cell lines to Pneumocystis infection. The authors addressed some of the key issues in the field currently and their review is relevant to the journal. Some of the strengths of this review include:

1.    A well summarized graphical presentation of the Pneumocystis-mediated lung epithelium immune responses and the signaling pathways involved.

2.     A well elaborated description of the interactions between alveolar type 1 and 2 cells with Pneumocystis supported by credible published work.

The review sheds light on how to solve the biggest challenge in the field of failing to culture Pneumocystis spp. in vitro/ ex vivo. There are a few minor concerns:

1.      The article stated in table 1 that the source of the 1HAEo- was the airway epithelial, with a DF508 mutation in CFTR gene, but according to the supplier’s description: It is unknown whether the donor had cystic fibrosis or not, however the cells do not contain the DF508 mutation in the CFTR gene or other CFTR mutations presenting 98% of all CF mutations” 1HAEo- Human Airway Epithelial Cell Line | SCC152 (emdmillipore.com). Thus, it would be important for the authors to mention or explain this discrepancy. Furthermore, it would be important to mention if these 1HAEo- cells would respond the same as non-CF airway epithelial cells.

2.     In the sections on AT1 and AT2 cells it would be worthwhile to state what makers were used to verify these cells in culture.

3.     The field now uses Club cells in lieu of Clara cells.    

Author Response

Reviewer #1:

The review sheds light on how to solve the biggest challenge in the field of failing to culture Pneumocystis spp. in vitro/ ex vivo. There are a few minor concerns:

  1. The article stated in table 1 that the source of the 1HAEo- was the airway epithelial, with a DF508 mutation in CFTR gene, but according to the supplier’s description: “It is unknown whether the donor had cystic fibrosis or not, however the cells do not contain the DF508 mutation in the CFTR gene or other CFTR mutations presenting 98% of all CF mutations” 1HAEo- Human Airway Epithelial Cell Line | SCC152 (emdmillipore.com). Thus, it would be important for the authors to mention or explain this discrepancy. Furthermore, it would be important to mention if these 1HAEo- cells would respond the same as non-CF airway epithelial cells.

The authors appreciate the comments. We have not removed the mention of the DF508 mutation in the table (line 614-615). We have also mentioned in lines 296-297 that the current state of 1Haeo- cytokine response to normal non-immortalized cells is unknown.

  1. In the sections on AT1 and AT2 cells it would be worthwhile to state what makers were used to verify these cells in culture.

      The authors again appreciate the comments. Going sequentially in the order of the manuscript and how the section on AT1 and AT2 were discussed.

AT1 discussion:

Lanken et al.: electron microscope study with no mention of cell markers.

Yoshida et al.: electron microscope study with no mention of cell markers.

Shiota et al.: electron microscope study with no mention of cell markers.

Beck et al.: No mention of AT1 cell marker sin this study but references two papers.

Hahn and Evans et. al: No mention of specific cell markers in manuscript but mention isolating type 1 cells by Dobbs protocol "An improved method for isolating type II cells in high yield and purity" 1986.

AT2 discussion:

Yoshida et al.: electron microscope study with no mention of cell markers.

Long et al.: electron microscope study with no mention of cell markers.

Pesanti: No mention of AT2 markers. Only mentions references for isolation of AT2 cells and macrophages.

Wang et al.: AT2 cell purity was mentioned and used " modified Papanicolaou staining and intracellular staining for pro-SPC". This was added to the manuscript  (lines 166-167).

Wang et al.: AT2 cell purity was mentioned and used "intracellular staining for pro-SPC". This was added to the manuscript  (lines 166).

  1. The field now uses Club cells in lieu of Clara cells.

The authors appreciate the clarification request, and now have included it in the manuscript throughout the Clara (club) section and added an appropriate reference.  

Reviewer 2 Report

The manuscript from Kottom, Carmona, and Limper details the response of lung epithelial cells to the fungal pathogen Pneumocystis.  The paper is slated to be an “Opinion”  but this is more of a review than an opinion.  The paper is organized by epithelial cell type in the lungs then by epithelial cell lines.  The references are broad and based on what has been published by the Pneumocystis community over the years.  Though the information is useful for the community, the writing is difficult to follow in spots and there are some places where it is difficult to tell what the point is that the authors are trying to make.  The model  at the end of the manuscript is useful but the epitopes need arrows or something to make it clear.  Specific comments are as follows:

1.       Line 84-86 summarizes a paper that uses conditional knockout mice as a model system for understanding clearance of Pneumocystis from the lungs in the absence of IKK2 in lung epithelium.  The sentence is difficult to understand as the conditional knockout mice are described oddly and there is not mention of mice in the sentence.  This is an example of the oddly worded sentences found throughout the manuscript.

2.       The authors mention the use of transwells in lines 99-102 and conclude that AT2 cells mount a host immune response to the organisms but don’t explain the experiment so it isn’t clear how the organisms die.  They don’t normally live long in culture. Were they in contact with the AT2 cells or separated by the transwell?  Either way, what killed them?

3.       Line 122-124 talks about Clara cells but it isn’t clear if the experiments are in vivo or in vitro.

4.       Line 142 talks about fibronectin production increasing having to do with proliferation of the organisms, but there is not explanation as to what fibronectin might have to do with organism proliferation.

5.       Line 145-146 says that there was an increase in IL-8 production but it wasn’t significant.  If the statistics says it isn’t significant, then there is no reason to mention this.

6.       Line 171-174 summarizes the obtaining of transcriptional data from experiments in which the life forms of Pneumocystis were separated but the authors don’t make any statement regarding what genes were different when epithelial cells were exposed to the life forms nor if anything important was concluded from this study.

7.       The conclusions would be more useful if the authors pointed out where the holes in the literature are and what approaches might be made to address them.

This document is full of English errors including problems with consistency of tense, missing words, misspelling, phrasing that doesn't make sense, and sentences that are too long.  Since the authors speak fluent English, this is basically an editing issue.  

Author Response

Reviewer #2:

The manuscript from Kottom, Carmona, and Limper details the response of lung epithelial cells to the fungal pathogen Pneumocystis.  The paper is slated to be an “Opinion”  but this is more of a review than an opinion.  The paper is organized by epithelial cell type in the lungs then by epithelial cell lines.  The references are broad and based on what has been published by the Pneumocystis community over the years.  Though the information is useful for the community, the writing is difficult to follow in spots and there are some places where it is difficult to tell what the point is that the authors are trying to make.  The model  at the end of the manuscript is useful but the epitopes need arrows or something to make it clear.

The authors are thankful for the comments. As requested by the editor(s), we have submitted this manuscript to their editing services for revision and included that version here. This manuscript was intended to be submitted as a "review" paper but did not meet the threshold word count so was recommended by the editor  to submit as an "opinion" piece. We have also updated Figure 1 to clearly mark the Pneumocystis epitopes as requested. We have addressed the specific reviewers’ comments below.

Specific comments are as follows:

  1. Line 84-86 summarizes a paper that uses conditional knockout mice as a model system for understanding clearance of Pneumocystisfrom the lungs in the absence of IKK2 in lung epithelium.  The sentence is difficult to understand as the conditional knockout mice are described oddly and there is not mention of mice in the sentence.  This is an example of the oddly worded sentences found throughout the manuscript.

The authors appreciate the comment. The authors have now restated that sentence and the sentence edited by the publishers editing service to hopefully make it better to understand to the reviewer and reader (lines 131-136).

  1. The authors mention the use of transwells in lines 99-102 and conclude that AT2 cells mount a host immune response to the organisms but don’t explain the experiment so it isn’t clear how the organisms die.  They don’t normally live long in culture. Were they in contact with the AT2 cells or separated by the transwell?  Either way, what killed them?

      The authors appreciate the comments on this section. In the original text, we did mention that the organisms were separated from the AT2 cells by transwells. We have expanded the details of the expt. and listed what the author thought killed to the fungal organism and also re-wrote these sentences to make them better to understand along with the publishers editing services (lines 146-163).

  1. Line 122-124 talks about Clara cells but it isn’t clear if the experiments are in vivo or in vitro.

      We have now added "in vivo" at the start of this sentence to clarify the experiment (line 179).

  1. Line 142 talks about fibronectin production increasing having to do with proliferation of the organisms, but there is not explanation as to what fibronectin might have to do with organism proliferation.

      The authors thank you for this comment. We have now added a sentence on what the authors hypothesis the role of fibronectin in the proliferative response (lines 218-220).

  1. Line 145-146 says that there was an increase in IL-8 production but it wasn’t significant.  If the statistics says it isn’t significant, then there is no reason to mention this.

      The authors have removed this sentence from the manuscript as requested.

  1. Line 171-174 summarizes the obtaining of transcriptional data from experiments in which the life forms of Pneumocystis were separated but the authors don’t make any statement regarding what genes were different when epithelial cells were exposed to the life forms nor if anything important was concluded from this study.

The authors appreciate the reviewers’ comments. We have now added which specific life form transcripts to the manuscript and also potential hypothesis (lines 262-277).

  1. The conclusions would be more useful if the authors pointed out where the holes in the literature are and what approaches might be made to address them.

      The authors appreciate the comments and have now added a sentence to the conclusion where current technology may be implemented and beneficial to understanding host lung epithelial/Pneumocystis interactions.

      Comments on the Quality of English Language

This document is full of English errors including problems with consistency of tense, missing words, misspelling, phrasing that doesn't make sense, and sentences that are too long.  Since the authors speak fluent English, this is basically an editing issue.

As requested by the editor, we have used the journal's editing service and have had the complete manuscript edited with this service. We have not included this version of the manuscript. 

Round 2

Reviewer 1 Report

n/a

Author Response

No additional comments/updates required by this reviewer. Thanks.

Reviewer 2 Report

This reads much better after editing.

The first sentence of the Introduction states "Pneumocystis are fungus...".  Is this a proper tense?  It just sounds wrong.

It was my understanding that the term "Clara" cell has been changed to "club" cell because of the antisemitic politics he held to and his engagement in the Third Reich.  Given that, I'd recommend that rather than put "club" in parentheses, you call them club cells and put something like (formerly called Clara cells) in parentheses.  Do this once at first use (section 2) and then just go with club cells.

Author Response

As requested, the authors changed the tense of the first sentence.

Point #2, the authors changed the Clara/club cell request and included the reviewers citation. Thanks.